# Stillbirth incidence and determinants in a tertiary health facility in the Volta Region of Ghana

**Anthony Kwame Dah**[1,2], **Joseph Osarfo**[3]*, **Gifty Dufie Ampofo**[3], **Adu Appiah-Kubi**[1,2], **Hintermann Mbroh**[2], **Wisdom Klutse Azanu**[1,2], **Afia Tabuaa Sakyi**[2], **Lydia Abradu**[2], **Emmanuel Senanu Komla Morhe**[1,2]

**1** Department of Obstetrics and Gynaecology, School of Medicine, University of Health and Allied Sciences, Ho, Ghana, **2** Directorate of Obstetrics and Gynaecology, Ho Teaching Hospital, Ho, Ghana, **3** Department of Community Health, School of Medicine, University of Health and Allied Sciences, Ho, Ghana

* josarfo@uhas.edu.gh

**Data Availability Statement:** All relevant data are within the paper and its Supporting information files.

## Abstract

### Background

Stillbirths are indicators of the quality of obstetrics care in health systems. Stillbirth rates and their associating factors vary by socio-economic and geographical settings. Published data on stillbirths and their associating factors in the Volta Region of Ghana are limited. This limits understanding of local factors that must be considered in designing appropriate interventions to mitigate the occurrence of stillbirths. This study determined the incidence of stillbirths and associated factors among deliveries at Ho Teaching Hospital (HTH) and contributes to understanding the consistent high stillbirths in the country and potentially in other low-resourced settings in sub-Saharan Africa.

### Method

This was a prospective cohort study involving pregnant women admitted for delivery at HTH between October 2019 and March 2020. Data on socio-demographic characteristics such as age and employment, obstetric factors including gestational age at delivery and delivery outcomes like birthweight were collected using a pretested structured questionnaire. The primary outcome was the incidence of stillbirths at the facility. Summary statistics were reported as frequencies, percentages and means. Logistic regression methods were used to assess for association between stillbirths and independent variables including age and birthweight. Odds ratios were reported with 95% confidence intervals and associations with p-values < 0.05 were considered statistically significant.

### Results

A total of 687 women and their 702 newborns contributed data for analysis. The mean age (SD) was 29.3 (6.3) years and close to two-thirds had had at least one delivery previously. Overall stillbirth incidence was 31.3 per 1000 births. Of the 22 stillbirths, 17 were antepartum. Pre-eclampsia was the most common hypertensive disorder of pregnancy observed

**Funding:** The authors received no specific funding for this work.

**Competing interests:** The authors have declared that no competing interests exist.

(49.3%, 33/67). Among others, less than 3 antenatal visits and low birthweight increased the odds of stillbirths in the bivariate analysis. In the final multivariate model, pregnancy and delivery at 28–34 weeks gestation [AOR 9.37(95% CI 1.18–74.53); p = 0.034] and induction of labour [AOR 11.06 (95% CI 3.10–39.42); p < 0.001] remained significantly associated with stillbirths.

## Conclusion

Stillbirth incidence was 31.3 per 1000 births with more than half being antepartum stillbirths. Pregnancy/delivery at 28–34 weeks' gestation increased the odds of a stillbirth. Improving the quality of antenatal services, ensuring adherence to evidence-based protocols, accurate and prompt diagnosis and timely interventions of medical conditions in pregnancy particularly at 28–34 weeks' gestation could reduce incidence of stillbirths.

## Introduction

Adverse pregnancy outcomes such as stillbirths mar the joyous expectations of women, their families and societies at large. Stillbirth rates have declined globally over the years from 24.7 per 1000 births in 2000 to 13.9 per 1000 births in 2019 with a global annual decline rate of 2.3% [1, 2] About 77% of the global burden of stillbirths occur in developing countries with an estimated pooled stillbirth rate of 21.3 per 1000 births for Africa in 2016 and an annual declining rate of 3% [2, 3]. Reported stillbirth rates for sub-Saharan African countries vary considerably with rates ranging from 20.3 to 118.1 per 1000 births across Malawi, Zimbabwe, Kenya and Sierra Leone over 2014 /2015 [4] and a more recent rate of 61.2 per 1000 births in Nigeria over 2019/2020 [5]. Even within countries, the incidence of stillbirths varies with different geographical and resource settings. In Nigeria, stillbirth rates of 56.1 per 1000 births and 104.6 per 1000 births were recorded in low resource settings in South-East and Kano respectively [6]. Also, stillbirth rates of 87 and 67 per 1000 births were reported in the Southern and Eastern Ethiopia respectively [7, 8].

Lower antepartum stillbirth rates, compared with intrapartum stillbirths, have been reported widely including sub-Saharan Africa [9–11]. The reported risk factors for stillbirths include maternal ages < 20–24 years and ≥35 year, preterm births, congenital anomalies, low birth weight, referral from another hospital and pregnant women with no previous birth experience [3, 12, 13]. Maternal age of 35 years or more has about twice the odds of a stillbirth while preterm birth has 20 times the odds of stillbirths [13]. Other studies have noted poor quality of antenatal care (ANC), hypertensive disorders of pregnancy, low birth weight (LBW) hypoxic intrapartum deaths, antepartum haemorrhage, and perinatal infection, prolonged labour for more than 12 hours as important risk factors that increase the odds of stillbirths [12, 14–17].

In Ghana, stillbirth rates ranging between 27 and 105.6 per 1000 births have been reported across a number of regions over 2013/2015 [15, 17–19]. The current national stillbirth rate of 20 per 1000 births was reported in 2017 in the last maternal and health survey (GHMS 2017). Contrary to earlier reports [9–11], a higher proportion of antepartum stillbirths, relative to the intrapartum variety, was reported in the Greater Accra Region of Ghana [20]. Various studies in Ghana have identified hypertensive disorders of pregnancy, LBW, hypoxic intrapartum deaths, antepartum haemorrhage, and perinatal infection, prolonged labour more than 12 hours and macrosomic neonates as important risk factors for stillbirth in the country [12, 15–

17]. However, published data on stillbirth rates in the Volta Region are limited and relatively little is known about local contextual factors that impact on stillbirths and around which targeted interventions can be built. Literature review showed only one retrospective study conducted in the Hohoe Municipal Hospital (now Volta Region regional hospital) [17] and which seemingly assessed only low birthweight as a risk factor for stillbirths. Though this was a large study with data for 4262 mother-infant pairs, it was restricted to singleton pregnancies and this may have resulted in an underestimation of the stillbirth prevalence of 27 per 1000 deliveries reported [17]. In addition, the study utilized routine health data collected over 2013–2014 and these may have had appreciable errors in their documentation. It is therefore pertinent to assess current stillbirth rates and associating risk factors in a prospective study to enable health managers and policy makers evaluate the effectiveness of existing interventions including the quality of obstetric care and the need for new interventions based on newly found dynamics.

## Materials and methods

### Study site description, study design and study population

A prospective cohort study was conducted at HTH between 1st October, 2019 and 31st March, 2020. The study population were women admitted for labour and delivery or prelabour caesarean section and their new born babies at HTH within the study period. The HTH is the fifth and youngest public teaching hospital in Ghana and serves as a referral center for the Volta Region and also parts of the Eastern region. The hospital provides services to clients from the Togo as well. Approximately 1,800 to 2,000 deliveries are conducted annually at the facility (Biostatistics Unit, HTH, 2022). The hospital has a 38-bed capacity maternity ward, a labour ward and an adjacent obstetric theatre. All antenatal cases of gestation 28 weeks and above presenting with complications, those for elective caesarean delivery as well as all post-delivery cases are admitted to the maternity ward. Over the period of data collection, the maternity ward and antenatal clinic were run by 30 midwives, 6 house-officers and 4 obstetrician/gynaecologists.

### Sample size determination

There was no formal sample size estimation for this study. All women admitted for labour and delivery or prelabour caesarean section and their new born babies, over the study period, were assessed for eligibility and included if found eligible and if consent was given. In all, 696 women (with their 714 newborns) were assessed for study inclusion. Of these, 687 women (and 702 newborns) were found eligible and included in the study.

### Inclusion and exclusion criteria

Women, referred and non-referred, admitted for labour and delivery or for prelabour caesarean section at HTH within the study period together with their new born babies were eligible to participate in the study. The women referred with intrauterine fetal deaths (IUFD) and those that delivered live or dead babies before 28 completed weeks of gestation (abortions) at HTH were excluded.

### Data collection tool and procedures

A structured questionnaire, developed by the investigators from literature review, was used to collect sociodemographic data through face-to-face interviews. Obstetric history of the current pregnancy including antenatal care clinic contacts, medical conditions such as hypertensive disorders of pregnancy and diabetes in pregnancy, gestational age in completed weeks at

delivery, birth status of the neonate at delivery and birth weight were extracted from maternal and child health record books, admission registers and delivery registers. Three registered midwives were recruited and trained for two days on questionnaire administration and data collection procedures by the lead author who is an obstetrician/gynaecologist. The training also emphasized translation from English to the local Ewe and Twi languages for uniformity. The questionnaire was administered in English, Ewe or Twi depending on the respondent's preference following consent. For multiple pregnancies, data was collected on each birth separately with a unique identity number. Participants were recruited daily and consecutively as they were admitted for labour and delivery or for prelabour caesarean operation. The questionnaire was pretested among ten pregnant women admitted for labour and delivery at HTH in August 2019 and relevant changes made to its structure before using it.

## Data management and analysis

The questionnaires were checked for completeness and accuracy. Data were double entered in Microsoft Excel, cleaned and exported into Statistical Package for Social Sciences (SPSS) version 22 (IBM Corp., Armonk, NY, USA) for analysis. Descriptive analyses were performed and presented as frequencies, percentages, means and standard deviation. The primary study outcome was the incidence of stillbirths at HTH. The secondary outcomes included the proportion of stillbirths among women referred with live fetuses from other health facilities to HTH for labour and delivery or prelabour caesarean operation, the proportion of stillbirths among women with hypertensive disorders of pregnancy and the types of hypertensive disorders of pregnancy among the respondents. Bivariate logistic regression analysis was used to assess for associations between stillbirths and the independent variables including participant's socio-demographic characteristics and obstetric factors of the index pregnancy. A backward stepwise regression with $p < 0.1$ was done in the multivariate analysis. Odds ratios and their 95% confidence intervals were reported. An association in the final multivariate logistic regression model was considered statistically significant if $p < 0.05$.

## Definition of variables

A stillbirth is defined as a neonate born after 28 completed weeks of gestation with no spontaneous breathing or heart beat [21].

Antepartum (macerated) stillbirth implied the fetus died before onset of labour or delivery process was initiated, and intrapartum (fresh) stillbirth meant the fetus died during delivery process.

Index pregnancy was the pregnancy for which the woman was admitted and delivered during the study period.

Preterm birth was a delivery after 28 completed weeks but before 37 completed weeks of gestation (early preterm 28 to 34 weeks, and late preterm 35 to 36 completed weeks' gestation).

A baby was said to have LBW if it was born after 28 completed weeks of gestation and weighed less than 2.5kg at birth (extremely LBW less than 1kg, very LBW from 1kg to 1.499kg, and moderately LBW from 1.5 to 2.499kg).

Anaemia in pregnancy was defined as last haemoglobin check of less than 11g/dl before birth (as recorded in participant's maternal and child health records book).

ANC contacts (visits) of three or less was considered inadequate and four or more as adequate.

Also intermittent preventive treatment of malaria in pregnancy (IPTp) doses of two or less was considered inadequate and three or more as adequate.

## Ethical considerations

Ethical approval for the study was granted by the University of Health and Allied Sciences Research and Ethics Committee (protocol ID UHAS-REC A12 [7] 18–19). Permission to collect data from the hospital was also obtained from the management of HTH. Written informed consent was obtained from the study women. For those who were less than 18 years old, assent was obtained from the participant herself and written informed consent obtained from the participant's caregiver /guardian. The purpose of the study was explained to the women. They were assured that participation was voluntary and that refusal to participate was not going to result in denial of care or any other punitive measure. It was explained that they could withdraw from the study even after they had initially agreed to participate. The study assessments did not involve any invasive procedure and study identification numbers were used to anonymize the participants.

## Results

### Participants' background characteristics

Of 696 pregnant women assessed for inclusion, 9 were excluded (5 were referred with IUFD and 4 delivered before 28 weeks gestation). Six hundred and eighty-seven (687) women with their 702 newborns thus participated in the study and contributed data for analysis. Table 1 summarizes the study women's background characteristics. The youngest parturient was 13 years and the oldest was 46 years. The mean age (SD) was 29.3 (6.3) years. More than half of the women, 57.3% (402), were aged between 25 years and 34 years. At least 92.3% (634) had basic education. Majority of the women, 73.1% (502), were married and nine in ten resided in urban/peri-urban areas. Women with no previous delivery experiences formed a third, 33.6% (231), of the respondents.

Table 2 summarizes obstetric factors relating to the current pregnancy as well as characteristics of the resultant newborns. Six hundred and nineteen women (90.1%) had adequate ANC contacts while 69.6% (478) took adequate doses of IPTp. Among the study women, 51.7% (355) had anaemia and 1.9% (13) had diabetes in pregnancy. About 10% (69/687) of the babies were born preterm and premature rupture of fetal membranes (PROM) occurred in 5.7% (39) of the study women. Labour was induced in 10.6% (73) of the pregnancies. Ninety-six (13.7%) of the newborns had low birthweight while less than 1% (5) had birth trauma.

### Incidence of stillbirths

Of a total 702 deliveries, there were 22 stillbirths; giving an incidence of 31.3 per 1000 births. Seventeen (17) of these stillbirths were antepartum. Also, of the 22 stillbirths, 12, 2 and 8 were within 28–34, 35–36 and ≥ 37 completed weeks of gestation. Among the 17 antepartum stillbirths, 9 occurred within the gestational age range of 28–34 weeks. Three out of the 5 intrapartum stillbirths occurred within this gestational age range as well. All but one of the intrapartum stillbirths were delivered by caesarean operation with the following indications; (i) two previous caesarean section with severe pre-eclampsia at 31 weeks 4 days' gestation, (ii) one previous caesarean section with antepartum haemorrhage at 34 weeks' gestation, (iii) footling breech at 36 weeks' gestation and (iv) leading twin with cord prolapse at 37 weeks' gestation. The fifth intrapartum stillbirth was a preterm vaginal delivery at 32 weeks' gestation.

Of the 22 stillbirths at the facility over the study period, 6 were among referred women. Among referrals with live fetuses for labour and delivery or prelabour caesarean operation, the incidence of stillbirth was 43 per 1000 births (6/140). Eight (8) of all the stillbirth births over the study period were among women with hypertensive disorders of pregnancy and this gave a

**Table 1. Demographic characteristics of mothers included in the determination of stillbirth incidence at Ho Teaching Hospital.**

| Variable | Frequency (N = 687) | Percent (%) |
|---|---|---|
| **Age in years** | | |
| < 25 | 153 | 21.8 |
| 25–34 | 402 | 57.3 |
| ≥ 35 | 147 | 20.9 |
| **Formal education** | | |
| No | 53 | 7.7 |
| [a]Basic school | 261 | 38.0 |
| Senior high school | 139 | 20.2 |
| Tertiary | 234 | 34.1 |
| **Employment** | | |
| Unemployed | 118 | 17.2 |
| [b]Informal | 359 | 52.3 |
| [c]Formal | 210 | 30.5 |
| **Marital status** | | |
| Single | 173 | 25.2 |
| Married | 502 | 73.1 |
| No response | 12 | 1.7 |
| **Place of residence** | | |
| Rural area | 69 | 10.0 |
| Urban/peri-urban | 618 | 90.0 |
| **Previous delivery** | | |
| No | 231 | 33.6 |
| One or more | 443 | 64.5 |
| No response | 13 | 1.9 |

[a]Basic school refers to education up to junior high school.

[b]Informal employment refers to traders, farmers, hairdressers and other artisans etc.

[c]Formal employment refers to occupations like civil/ public servants and corporate workers

stillbirth incidence of 114.3 per 1000 births (8/70) in this specific sub-population. It must be noted that being referred and having a hypertensive disorder of pregnancy were not mutually exclusive as 30 of the referrals also had a hypertensive disorder of pregnancy.

## Types of hypertensive disorders of pregnancy observed

Nearly a tenth, 9.8% (67/687), of the women had hypertensive disorders of pregnancy and these included pre-eclampsia 49.3% (33/67), gestational hypertension 28.4% (19/67), eclampsia 10.4% (7/67), chronic hypertension with superimposed pre-eclampsia, 7.5% (5/67) and chronic hypertension in pregnancy 4.5% (3/67).

## Risk factors of stillbirths

In the bivariate analysis, none of the participant socio-demographic characteristics was associated with stillbirths (see Table 3). Women with inadequate ANC contacts had about 5 times the odds of having a stillbirth compared to those who had adequate ANC contacts [OR 4.92 (95% CI 1.84–13.15); p = 0.002]. Similarly, participants who had received less than 3 doses of IPTp had close to 4 times the odds of a stillbirth compared to those who received ≥3 doses

**Table 2. Obstetric factors of index pregnancy and characteristics of newborns assessed in the study.**

| Variable | Frequency (N = 687) | Percent (%) |
|---|---|---|
| **Fetus(es) in a pregnancy** | | |
| Twin | 15 | 2.2 |
| Singleton | 672 | 97.8 |
| **ANC contact** | | |
| Zero | 3 | 0.4 |
| 1–3 | 50 | 7.3 |
| ≥ 4 | 619 | 90.1 |
| No response | 15 | 2.2 |
| **IPTp dose** | | |
| Zero | 27 | 3.9 |
| 1–2 | 132 | 19.2 |
| ≥ 3 | 478 | 69.6 |
| No response | 50 | 7.3 |
| **Anaemia in pregnancy** | | |
| Yes | 355 | 51.7 |
| No | 311 | 45.3 |
| No response | 21 | 3.0 |
| *****SCD in pregnancy** | | |
| Yes | 15 | 2.2 |
| No | 621 | 90.4 |
| No response | 51 | 7.4 |
| **Diabetes in pregnancy** | | |
| Yes | 13 | 1.9 |
| No | 666 | 96.9 |
| No response | 8 | 1.1 |
| **Referral status** | | |
| Referred | 139 | 20.2 |
| Not referred | 548 | 79.8 |
| **&Gestation** | | |
| 28–34 | 44 | 6.4 |
| 35–36 | 25 | 3.6 |
| ≥ 37 | 618 | 90.0 |
| **@PROM** | | |
| Yes | 39 | 5.7 |
| No | 648 | 94.3 |
| **Labour** | | |
| Induced | 73 | 10.6 |
| Not induced | 614 | 89.4 |
| **Mode of delivery (N = 702)** | | |
| Caesarean | 218 | 31.1 |
| Vaginal | 484 | 68.9 |
| **Birth weight (Kg) (N = 702)** | | |
| ≤ 1.499 | 32 | 4.6 |
| 1.5–2.499 | 64 | 9.1 |
| ≥ 2.500 | 606 | 86.3 |
| **Congenital anomaly (N = 702)** | | |
| Yes | 11 | 1.6 |

(*Continued*)

**Table 2.** (Continued)

| Variable | Frequency (N = 687) | Percent (%) |
|---|---|---|
| No | 691 | 98.4 |
| **Birth trauma (N = 702)** | | |
| Yes | 5 | 0.7 |
| No | 697 | 99.3 |

*SCD in pregnancy refers to sickle cell disease in pregnancy.

&Gestation refers to gestational age in completed weeks at time of delivery.

@PROM refers to premature rupture of fetal membranes.

[OR 3.49 (95% CI 1.40–8.69); p = 0.007]. Study women with any of the hypertensive disorders of pregnancy had nearly 6 times the odds of a stillbirth relative to women without the condition [OR 5.56 (95% CI 2.25–13.77); p<0.001]. Preterm pregnancies or births showed increased odds of stillbirths with early preterm having markedly increased odds compared to late preterms (see Table 3). Labour induction also increased the odds of stillbirths by about six times (see Table 3). There were no records of stillbirths among the neonates born to women with sickle cell disease and diabetes in pregnancy.

In the multivariate analysis, only preterm pregnancies at 28–34 weeks' gestation [AOR 9.37 (95% CI 1.18–74.53) p = 0.034] and labour induction [AOR 11.06 (95% CI 3.10–39.42) p<0.001] remained significantly associated with stillbirths (see Table 3).

## Discussion

The study assessed the incidence and factors influencing stillbirths among deliveries at Ho Teaching Hospital. An overall stillbirth incidence rate of 31.3 per 1000 births was observed in the study with preterm delivery at 28–34 weeks' gestation and labour induction as key risk factors. Among pregnancies referred to HTH for delivery, the stillbirth incidence was 43 per 1000 births while among women with hypertensive disorders of pregnancy, it was 114 per 1000 births. The prevalence of hypertensive disorders of pregnancy was 9.8% and preeclampsia and chronic hypertension in pregnancy were the most and the least common forms respectively. To the best of the authors' knowledge, this is the first study reporting on stillbirth incidence and a comprehensive array of risk factors in the Volta Region. The study provides vital baseline information for action to alleviate the burden of stillbirths in HTH and possibly the rest of the Volta Region.

The stillbirth rate of 31.3 per 1000 births in this study falls within the stillbirth trend contained in the facility annual reports, ranging from 21.3/1000 births in 2015 to 35.3/1000 births in 2022 (Biostatistics Unit, HTH, 2015 to 2022). Also consistent with the facility annual reports is the finding that there were more antepartum stillbirths than intrapartum ones. The stillbirth incidence observed in the present study is slightly lower than the 34 per 1000 births in Navrongo, [18], the 59 per 1000 births recorded in Kumasi [15] and much lower than the 105.6 per 1000 births reported in Accra [20] all in Ghana. The HTH stillbirth rate is also lower than the rates of 39.6 per 1000 births and the 74 per 1000 births reported in Nigeria [22, 23].

The current study and those by Adu-Bonsaffoh et al. [20] and Dassah et al. [15] were conducted in Ghana's public tertiary health facilities; HTH in Volta Region, Korle-Bu Teaching Hospital (KBTH) in Greater Accra Region and Komfo-Anokye Teaching Hospital (KATH) in Ashanti Region. Differences in the stillbirth rates could be partly attributed to variations in the study designs, study populations, the study times and the population density of the study

**Table 3. Bivariate and multivariate logistic regression analysis output for association between stillbirth and independent variables.**

| Predictor Variable | Outcome Variable | | Crude OR | 95% CI | p-value | [5]AOR | 95% CI | p-value |
|---|---|---|---|---|---|---|---|---|
| | Stillbirth n(%) | Livebirth n(%) | | | | | | |
| **Age group (years)** | | | | | | | | |
| < 25 | 2 (1.3) | 151 (98.7) | 0.34 | 0.08–1.51 | 0.157 | | | |
| 25–34 | 15 (3.7) | 387 (96.3) | 1 | | | | | |
| ≥ 35 | 5 (3.4) | 142 (96.6) | 0.91 | 0.32–2.55 | 0.855 | | | |
| **Education** | | | | | | | | |
| No formal | 3 (5.6) | 51 (94.4) | 2.28 | 0.55–9.44 | 0.254 | | | |
| Basic school | 12 (4.5) | 255 (95.5) | 1.83 | 0.68–4.95 | 0.235 | | | |
| Senior high school | 1 (0.7) | 141 (99.3) | 0.28 | 0.03–2.31 | 0.235 | | | |
| Tertiary | 6 (2.5) | 233 (97.5) | 1 | | | | | |
| **Employment** | | | | | | | | |
| No | 5 (4.2) | 115 (95.8) | 1.82 | 0.52–6.41 | 0.353 | | | |
| Informal | 12 (3.3) | 356 (96.7) | 1.41 | 0.49–4.06 | 0.525 | | | |
| Formal | 5 (2.3) | 209 (97.7) | 1 | | | | | |
| **Marital status** | | | | | | | | |
| Single | 8 (4.5) | 169 (95.5) | 1.69 | 0.70–4.09 | 0.247 | | | |
| Married | 14 (2.7) | 499 (97.3) | 1 | | | | | |
| **Residence** | | | | | | | | |
| Rural | 4 (5.5) | 69 (94.5) | 1.97 | 0.65–5.98 | 0.233 | | | |
| Urban/peri-urban | 18 (2.9) | 611 (97.1) | 1 | | | | | |
| **Previous delivery** | | | | | | | | |
| No | 9 (3.8) | 226 (96.2) | 1.35 | 0.57–3.20 | 0.499 | | | |
| ≥ 1 | 13 (2.9) | 440 (97.1) | 1 | | | | | |
| **Fetus(es) in a pregnancy** | | | | | | | | |
| Twin | 1 (3.3) | 29 (96.7) | 1.07 | 0.14–8.22 | 0.949 | | | |
| Singleton | 21 (3.1) | 651 (96.9_ | 1 | | | | | |
| **Antenatal clinic visit** | | | | | | | | |
| ≤ 3 | 6 (11.3) | 47 (88.7) | 4.92 | 1.84–13.15 | 0.002 | 1.31 | 0.28–6.20 | 0.732 |
| ≥ 4 | 16 (2.5) | 616 (97.5) | 1 | | | 1 | | |
| **IPTp dose** | | | | | | | | |
| ≤ 2 | 10 (6.1) | 154 (93.9) | 3.49 | 1.40–8.69 | 0.007 | 0.55 | 0.16–1.89 | 0.340 |
| ≥3 | 9 (1.8) | 478 (98.2) | 1 | | | 1 | | |
| **Anaemia in pregnancy** | | | | | | | | |
| ≤ 10.9 g/dl | 9 (2.5) | 352 (97.5) | 0.60 | 0.25–1.43 | 0.250 | | | |
| ≥ 11.0 g/dl | 13 (4.1) | 306 (95.9) | 1 | | | | | |
| [1]HDP | | | | | | | | |
| Yes | 8 (11.4) | 62 (88.6) | 5.56 | 2.25–13.77 | <0.001 | 0.48 | 0.11–2.19 | 0.346 |
| No | 14 (2.2) | 618 (97.8) | 1 | | | 1 | | |
| **Referral status** | | | | | | | | |
| Referred | 6 (4.3) | 135 (95.7) | 1.48 | 0.57–3.86 | 0.419 | | | |
| Not referred | 16 (2.9) | 545 (97.1) | 1 | | | | | |
| [2]PROM | | | | | | | | |
| Yes | 1 (2.4) | 40 (97.6) | 0.72 | 0.09–5.49 | 0.951 | | | |
| No | 21 (3.4) | 605 (96.6) | 1 | | | | | |
| [3]Gestational age | | | | | | | | |
| 28–34 | 12 (25.5) | 35 (74.5) | 26.49 | 10.17–68.98 | <0.001 | 9.37 | 1.18–74.53 | 0.034 |
| 35–36 | 2 (6.9) | 27 (93.1) | 5.72 | 1.16–28.25 | 0.032 | 6.40 | 0.57–72.04 | 0.133 |

(*Continued*)

**Table 3.** (Continued)

| Predictor Variable | Outcome Variable | | Crude OR | 95% CI | p-value | [5]AOR | 95% CI | p-value |
|---|---|---|---|---|---|---|---|---|
| | Stillbirth n(%) | Livebirth n(%) | | | | | | |
| ≥ 37 | 8 (1.3) | 618 (98.7) | 1 | | | 1 | | |
| **Labour** | | | | | | | | |
| Induced | 8 (11.3) | 63 (88.7) | 6.41 | 2.56–16.05 | <0.001 | 11.06 | 3.10–39.42 | <0.001 |
| Not induced | 13 (2.1) | 617 (97.9) | 1 | | | 1 | | |
| **Mode of delivery** | | | | | | | | |
| Caesarean | 4 (1.8) | 214 (98.2) | 0.48 | 0.16–1.45 | 0.194 | | | |
| Vagina | 18 (3.7) | 466 (96.3) | 1 | | | | | |
| [4]**Birth weight in kg** | | | | | | | | |
| ≤ 1.499 | 10 (31.3) | 22 (68.7) | 38.90 | 13.54–111.76 | <0.001 | 8.51 | 0.91–79.18 | 0.060 |
| 1.5–2.499 | 5 (7.8) | 59 (92.2) | 7.25 | 2.23–23.56 | 0.001 | 1.73 | 0.19–15.42 | 0.625 |
| ≥ 2.500 | 7 (1.2) | 599 (98.8) | 1 | | | 1 | | |
| **Congenital anomaly** | | | | | | | | |
| Yes | 1 (9.1) | 10 (90.9) | 3.14 | 0.38–25.69 | 0.285 | | | |
| No | 21 (3.0) | 670 (97.0) | 1 | | | | | |

[1]HDP is hypertensive disorders of pregnancy

[2]PROM is premature rupture of membranes

[3]Gestational age is in completed weeks at birth. Recategorized as <37 and ≥37 weeks, it was observed that term pregnancies / babies had, at least, a 99% reduced odds of a stillbirth compared to preterms [OR 0.06 (95% CI 0.02–0.14); p < 0.001] in the bivariate analysis. In the multivariate analysis and with the recategorization, babies born at gestational age ≥37 had 88% reduced odds a stillbirth compared to those born at less than 37 completed weeks [AOR 0.12 (95% CI 0.02–0.84); p = 0.033]. The wider confidence intervals in the table result from the small sizes of the sub-groups

[4]With birth weight recategorized as <2.5Kg and ≥2.5Kg in the bivariate analysis, babies with birthweight at least 2.5Kg had a markedly reduced odds of a stillbirth [OR 0.06 (95% CI 0.03–0.16); p < 0.001]. In the multivariate analysis, however, this association was lost [AOR 3.08 (95%CI 0.44–21.33); p = 0.256].

[5]AOR is adjusted odds ratio. Each variable included for multivariate analysis was adjusted for the other variables with statistically significant association in the bivariate model

areas. Whilst the current study was prospective on all deliveries including twin births in 2019/ 2020, the Kumasi study was retrospective and involved only vaginal births in 2009 while the Greater Accra study was also retrospective but limited to singleton deliveries in 2015. Also, the population density within the catchment served and hence the pressure on HTH is expectedly much less compared to KATH in Kumasi and KBTH in Greater Accra. In addition, the maternal and child health interventions over the years resulting in global decline in stillbirths [1, 2] could partly explain the lower stillbirth rate observed in the present study though the HTH annual reports show an increasing trend in stillbirths between 2015 and 2022. A future study on stillbirths over a much longer period at HTH could give better clarity on the trend of stillbirth rates at the facility. The differences in the study sites in terms of socio-cultural, geographical and political settings, as well as population density within the facility catchment area could explain the variation in stillbirth rates between the current study in Ghana and the 39.6 and 74 per 1000 births reported in Nigeria [22, 23].

The HTH stillbirth rate of 31.3 per 1000 births is higher than the 27 per 1000 births recorded in Hohoe Municipality of the Volta Region, the 23 per 1000 births in the Upper East Region and the current Ghana national stillbirth rate of 20 per 1000 births [17, 24, 25] but compares well with the 29.7 per 1000 births in a tertiary care hospital in South India [26]. Similarly, the stillbirth rate in the present study is higher than the pooled stillbirth rate for Africa of 21.3 per 1000 births [3] and the global stillbirths estimate of 13.9 per 1000 births for 2019 [1].

The current study and that in Hohoe [17] were both conducted in urban settlements in Ghana. However, the different study populations and designs could explain the observed differences in the reported stillbirth rates. The current study was prospective and involved all births while the study at Hohoe included only singletons. This makes the latter amenable to underestimation of the stillbirth rate.

The proportion of antepartum stillbirths (17 out of 22) was higher than intrapartum stillbirths in the present study. This compares favourably with a Ghanaian study which reported 59.6% of antepartum stillbirths [20] but is inconsistent with other study findings of higher proportions of intrapartum stillbirths, ranging 50.7%-67.4%, in sub-Saharan Africa and outside the continent [9–11]. The variations in proportions of antepartum and intrapartum stillbirths between Ghanaian health facilities and other health facilities outside Ghana could be attributed to the differences in the health care policies and implementation strategies. Recommended intrapartum management strategies practiced at the study site include 4-hourly review of all cases in labour by obstetrician-led teams of doctors, strict monitoring of labour using partographs, cardiotocograph monitoring of high-risk pregnancies in labour, and regular in-service training of midwives and doctors on interpretation of the cardiotocograph. These result in prompt identification of compromised fetuses and timely interventions with emergency caesarean sections being performed within 15–30 minutes of diagnosis to prevent stillbirth.

Though stillbirth rate of 43 per 1000 births occurred among referred pregnancies with live fetuses, referral from other health facilities to HTH for delivery was not associated with stillbirths contrary to previous study findings [22, 27]. This may arise from capacity building measures for health facilities in Ghana over the years including the posting of specialist obstetricians to some district-level health facilities in the country such as Ho Municipal Hospital, Sogakofe District Hospital and Abor District Hospital in the Volta Region. Also, there are effective collaborations, resulting from good interpersonal relationships and communication, between health care providers of HTH and the referring health facilities. Specialist obstetricians from HTH frequently visit the peripheral health facilities within the catchment areas including Oti Region. They provide surgical services and conduct maternal death audits with an eye to identifying gaps and missed opportunities that district-level health staff need to address in order to prevent such future occurrences. Often the women referred in pregnancy are from clinics, maternity homes and community-based health and planning services (CHPS) compounds that are not very far from HTH. The referrals from the district hospitals are fewer and mostly for intensive care unit (ICU) services, either in pregnancy or after delivery with complications. These could explain why there was no association between stillbirths and referred pregnancies with live fetuses for delivery at HTH in the current study.

Preterm pregnancy of gestation 28–34 completed weeks at time of delivery, as a determinant of stillbirth in the current study, needs cautious interpretation. More than half of antepartum stillbirths (52.9%) and intrapartum stillbirths (60%) occurred within the gestational age of 28–34 completed weeks. This suggests preterm pregnancy/birth at 28–34 completed weeks' gestation has elevated risk for both antepartum and intrapartum stillbirths respectively with invariable low birthweight. This is consistent with the findings from Hohoe and Kumasi in Ghana [15, 17] and sub-Saharan Africa [13].

Hypertensive disorders of pregnancy, one of the known risk factors for adverse preterm pregnancy outcomes [28] was present in 9.8% of the study women and the most common form was preeclampsia followed by gestational hypertension. This is consistent with the 8% pooled prevalence of hypertensive disorders of pregnancy with both preeclampsia and gestational hypertension being equally common in the sub-Saharan Africa [28]. Though there was no statistically significant association with stillbirth in the final multivariate model, other factors in the present study such as inadequate ANC clinic contacts and inadequate IPTp

administration are also commonly reported risk factors for adverse preterm pregnancy outcomes including stillbirths [13, 16, 20, 28, 29]. Therefore, measures to address antepartum stillbirths at the facility could include improving antenatal care quality and contacts, adequate doses of IPTp administration to all pregnant women and early detection and effective management of hypertensive disorders of pregnancy. High quality antenatal care services could minimize incidence of antepartum stillbirths from malaria, urinary tract infection, PROM, diabetes and hypertensive disorders of pregnancy [30, 31]. Evidence based antenatal care services and protocols on management of medical conditions in pregnancy with particular attention to preterm pregnancy at 28–34 weeks' gestation, adequate training of newly recruited staff and regular in-service training on the protocols could partly reduce the antepartum stillbirths. Regular review of high-risk pregnancies by obstetrician specialists, particularly in the preterm periods, without completely leaving the perceived low-krisk pregnancies in the hands of midwives and house-officers could enhance early detection and timely management of complications during pregnancy to minimize the antepartum stillbirths. Effective use of antepartum fetal surveillance tools in clearly spelt-out protocols at the facility could also contribute to reduction in antepartum stillbirths. Besides, early administration of tocolytics to abort premature uterine contractions [32], dexamethasone administration to mature fetal lungs [33] and close monitoring could also minimize preterm antepartum and intrapartum stillbirths.

The association between labour induction and stillbirth in the present study also needs cautious interpretation. Firstly, the very wide 95% confidence interval around the adjusted odds ratio in the final multivariate model suggests the number of participants studied for this independent variable was small and thus had reduced precision. Secondly, it was found that all but one of the intrapartum stillbirths were delivered through prelabour caesarean operation for medical conditions contraindicated to labour induction. This more or less 'discredits' labour induction as a risk factor for stillbirths, particularly intrapartum ones, in the facility. Similar to the practice in other settings [34], fetuses that die in-utero at the facility are preferably delivered through the vaginal route unless contraindicated. About half of the antepartum stillbirths were induced for vaginal delivery and this could underlie the apparent association of labour induction with stillbirths in the facility. While labour induction may have been responsible for reductions in stillbirths in national-level reports in England and Denmark [35, 36], they increased the adjusted odds of fresh stillbirths in Africa and Asia in a WHO report on maternal and infant health [37].

At HTH, labour induction is performed for low-risk pregnancies between 41 and 42 weeks' gestation with no contraindication to vaginal delivery based on existing evidence [38]. Indications for provider-initiated preterm delivery at the facility follow evidence-based protocols and recommendations such as WHO recommendations on interventions to improve preterm birth outcomes, the International Federation of Gynaecology and Obstetrics (FIGO) good practice recommendations on modifiable causes of iatrogenic preterm birth, the National Institute for Health and Care Excellence (NICE) Guidelines for preterm labour and birth and the local national safe motherhood protocols. Intrapartum stillbirth rate at the facility was addressed some years ago based on the perinatal mortality audit recommendations. The measures instituted included obstetrician-led team based management of all cases in labour, identification and classification of risk for every labour case, strict adherence to partograph monitoring of labour and prompt review and timely intervention of abnormal labour, regular training of midwives and doctors on interpretation of cardiotocograph and the use in monitoring selected labour based on evidence-based inclusion criteria. These could explain the low incidence of intrapartum stillbirths recorded in the current study. Also, as a result, emergency caesarean operations are currently offered within 15–30 minutes of diagnosis. Notwithstanding, four out of the five intrapartum stillbirths were caesarean deliveries, and a previous study

had ascribed elevated stillbirths to poor quality of care for a new born baby during caesarean operation [9]. Future studies could focus on examining the risk factors for antepartum stillbirths particularly among preterm pregnancies within the period 28–34 weeks' gestation and intrapartum stillbirths at the facility.

## Strength and limitations

This study is the first to report on stillbirths at HTH in the Ho Municipality. The study reported the incidence of stillbirths and identified pregnancy at 28–34 completed weeks' gestation and the birth as the leading determinants of stillbirths at the facility. However, the 95% CI around the odds for the gestational age, labour induction and birthweight were wide. While these may suggest a low level of precision for the associations with stillbirths and the need for cautious interpretation of these particular findings, it must be noted that the wide intervals arise mostly from the small sizes of the sub-categories under these variables. Furthermore, the study was done at a single site and the findings thereof may have limited external validity. Nevertheless, the study reports findings comparable to previous ones and provide fine scale data for targeted action to reduce stillbirths.

## Conclusion

An overall stillbirth incidence of 31.3 per 1000 births was observed among deliveries at HTH over the study period, and at least 75% of the occurring stillbirths were antepartum. Pregnancy at 28–34 completed weeks' gestation and the birth increased the odds of a stillbirth among the study participants. These factors need to be controlled through measures including adequate ANC contacts where medical conditions such as hypertensive disorders of pregnancy can be detected early and managed, and ensuring the women receive adequate doses of IPTp. Future multicenter studies are needed to help give more precise national perspectives to what could be the key determinants of stillbirths in the country. This will help formulate context-specific interventions and strengthen or refocus national-level policies to tackle the menace of stillbirths in the country.

## Supporting information

**S1 Checklist. Completed STROBE checklist.**
(DOCX)

**S1 Dataset. Stillbirths work dataset underlying the study findings.**
(XLSX)

## Acknowledgments

We are grateful to the study women for their time.

## Author Contributions

**Conceptualization:** Anthony Kwame Dah, Joseph Osarfo, Hintermann Mbroh, Emmanuel Senanu Komla Morhe.

**Data curation:** Anthony Kwame Dah, Joseph Osarfo, Adu Appiah-Kubi, Afia Tabuaa Sakyi, Lydia Abradu.

**Formal analysis:** Anthony Kwame Dah, Joseph Osarfo, Gifty Dufie Ampofo.

**Methodology:** Anthony Kwame Dah, Joseph Osarfo, Gifty Dufie Ampofo.

**Project administration:** Anthony Kwame Dah, Afia Tabuaa Sakyi, Lydia Abradu.

**Supervision:** Anthony Kwame Dah, Adu Appiah-Kubi, Hintermann Mbroh, Wisdom Klutse Azanu, Afia Tabuaa Sakyi.

**Validation:** Joseph Osarfo, Gifty Dufie Ampofo, Adu Appiah-Kubi, Wisdom Klutse Azanu, Afia Tabuaa Sakyi, Lydia Abradu.

**Writing – original draft:** Anthony Kwame Dah, Joseph Osarfo, Gifty Dufie Ampofo.

**Writing – review & editing:** Anthony Kwame Dah, Joseph Osarfo, Gifty Dufie Ampofo, Adu Appiah-Kubi, Hintermann Mbroh, Wisdom Klutse Azanu, Emmanuel Senanu Komla Morhe.

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
