## [Decision Letter · Decision Letter 0]

8 Nov 2023

PONE-D-23-17720Stillbirth incidence and determinants in a tertiary health facility in the Volta Region of Ghana.PLOS ONE

Dear Dr. Osarfo,

Thank you for submitting your manuscript to PLOS ONE. After careful consideration, we feel that it has merit but does not fully meet PLOS ONE’s publication criteria as it currently stands. Therefore, we invite you to submit a revised version of the manuscript that addresses the points raised during the review process.

We look forward to receiving your revised manuscript.

Kind regards,

Rornald Muhumuza Kananura, PhD

Academic Editor

PLOS ONE

Journal Requirements:

Reviewers' comments:

Reviewer's Responses to Questions

**Comments to the Author**

1. Is the manuscript technically sound, and do the data support the conclusions?

Reviewer #1: Yes

Reviewer #2: Yes

2. Has the statistical analysis been performed appropriately and rigorously? 

Reviewer #1: Yes

Reviewer #2: Yes

3. Have the authors made all data underlying the findings in their manuscript fully available?

Reviewer #1: Yes

Reviewer #2: Yes

4. Is the manuscript presented in an intelligible fashion and written in standard English?

Reviewer #1: Yes

Reviewer #2: Yes

5. Review Comments to the Author

Reviewer #1: Dear Authors

Thank you for a well written document. The manuscript discusses an important subject that aims to improve the provision of maternal and child health in this setting and proffer lessons that could be applied elsewhere. I have a few comments to make.

Abstract:

Authors need to include the total deliveries in the results. The use of percentage where n<30 is not recommended.

Main manuscript:

The introduction is well written giving a clear context to the reader.

Materials and methods:

Line 105 The authors should be consistent in the use of the comma i.e., 1,800 and 2000 deliveries.

How does seasonality affect the results? Could the results have been different if the study was conducted for a full year?

Line 153 to 159 How did the authors define univariate logistic regression analysis? Is this not a bivariate analysis after which they enter variables with a p-value less than 0.20 into the multivariate logistic regression model.

Results:

Table 1 and 2 The title is not informative. Should show who/what, where and when.

The use of percentages when n<30 is not recommended. Correct this throughout the manuscript.

Line 260 Is this univariate or bivariate analysis. Univariate analysis does not generate odds ratios.

Table 3 Include the values for the 2xn table for us to see where the CORs came from

Reviewer #2: Abstract: Results- Some basic demographic information would have helped to appreciate who the study population were. The univariate analysis is also clearly missing here

Sample size determination

Why was the sample size not determined?

Data collection tool and procedures

Was any validation done for this questionnaire based on the fact that there was translation from English to other languages?

6. PLOS authors have the option to publish the peer review history of their article (what does this mean?). If published, this will include your full peer review and any attached files.

Reviewer #1: No

Reviewer #2: **Yes: **Christian Obirikorang

---

## [Author Response · Author response to Decision Letter 0]

14 Nov 2023

RESPONSE TO REVIEWERS’ COMMENTS

REVIEWER #1

1. Authors need to include the total deliveries in the results (Abstract). The use of percentage where n<30 is not recommended.

Response: Both comments have been addressed. The first two sentences of the results section under the abstract now reads as;

“A total of 687 women and their 702 newborns contributed data for analysis. Overall stillbirth incidence was 31.3 per 1000 births. Of the 22 stillbirths, 17 were antepartum.”

2. The introduction is well written giving a clear context to the reader

Response: The authors are thankful to the reviewer for such kind and encouraging remarks. 

3. Line 105. The authors should be consistent in the use of the comma i.e., 1,800 and 2000 deliveries.

Response: This has been noted and addressed. The sentence in question now reads as;

“Approximately 1,800 to 2,000 deliveries are conducted annually at the facility (Biostatistics Unit, HTH, 2022).”

4. How does seasonality affect the results? Could the results have been different if the study was conducted for a full year?

Response: The results could have been different if the study had been conducted for a full year. However, the ‘seasonality’ element has been taken off.

5. Line 153 to 159 How did the authors define univariate logistic regression analysis? Is this not a bivariate analysis after which they enter variables with a p-value less than 0.20 into the multivariate logistic regression model.

Response: ‘Univariate’ was used to mean one exposure variable was under consideration at a time. However, the authors have changed to ‘Bivariate’ as suggested

‘Univariate’ has been changed to “Bivariate” throughout the manuscript. Covariates that were statistically significant at p < 0.1 were entered into multivariate analysis in a backward stepwise regression.

 Under Methods, Data Management and Analysis, the relevant sentence has been revised to read as;

“A backward stepwise regression with p<0.1 was done in the multivariate analysis”

6. Table 1 and 2....The title is not informative. Should show who/what, where and when.

Response: The authors agree with the reviewer on this and have reviewed the titles accordingly as shown below;

Table 1 is now titled as; “Demographic characteristics of mothers included in the determination of stillbirth incidence at Ho Teaching Hospital”

Table 2 is now titled as; “Obstetric factors of index pregnancy and characteristics of newborns assessed in the study”

The authors opine that these titles suffice and that the other elements pointed out can be teased from the body of the manuscript. We also wish to guard against excessively long titles.

7. The use of percentages when n<30 is not recommended. Correct this throughout the manuscript.

Response: The authors thank the reviewer for this comment as it was not known to us. The correction has been done throughout the manuscript

8. Line 260 Is this univariate or bivariate analysis. Univariate analysis does not generate odds ratios.

Response: It has been changed to ‘bivariate’

9. Table 3… Include the values for the 2xn table for us to see where the CORs came from

Response: Table 3 has been revised to include the values (and percentages) for the 2 x n table to show where the CORs came from as suggested.

REVIEWER #2

1. Abstract: Results- Some basic demographic information would have helped to appreciate who the study population were. The univariate analysis is also clearly missing here

Response: The suggested findings have been incorporated. The results (Abstract) now reads as;

“A total of 687 women and their 702 newborns contributed data for analysis. The mean age (SD) was 29.3 (6.3) years and close to two-thirds had had at least one delivery previously. Overall stillbirth incidence was 31.3 per 1000 births. Of the 22 stillbirths, 17 were antepartum. Pre-eclampsia was the most common hypertensive disorder of pregnancy observed (49.3%, 33/67). Among others, less than 3 antenatal visits and low birthweight increased the odds of stillbirths in the bivariate analysis. In the final multivariate model, pregnancy and delivery at 28–34 weeks gestation [AOR 9.37(95% CI 1.18 – 74.53); p=0.034] and induction of labour [AOR 11.06 (95% CI 3.10 – 39.42); p < 0.001] remained significantly associated with stillbirths.”

2. Why was the sample size not determined?

Response: There were very few studies and none at all in our regional setting reporting current stillbirth incidence. The only published study from our regional setting was in Hohoe and it reported stillbirth prevalence among singletons in a retrospective study. This baseline was not deemed appropriate to use in sample size determination. We emphasize ‘regional setting’ because we desired to be able to speak to our local context in this assessment.

Hence, we chose the permitted alternative of gathering data in a prospective study over a 6-month period to enable us report stillbirth incidence in our setting.

Secondly, this prospective study was meant to give a preliminary insight into the problem of stillbirth in the facility and the findings thereof will inform larger and more comprehensive future studies into stillbirths in our setting.

3. Was any validation done for this questionnaire based on the fact that there was translation from English to other languages?

Response: Pretesting the questionnaire was the only validation mechanism employed. Though the interviewers were Ewe and Twi speakers, they were still trained to standardize how the questions were framed during the interviews to minimize response bias and to improve internal validity. The pretesting of the questionnaire has been reported.

---

## [Decision Letter · Decision Letter 1]

6 Dec 2023

Stillbirth incidence and determinants in a tertiary health facility in the Volta Region of Ghana.

PONE-D-23-17720R1

Dear Dr. Osarfo,

We’re pleased to inform you that your manuscript has been judged scientifically suitable for publication and will be formally accepted for publication once it meets all outstanding technical requirements.

Kind regards,

Rornald Muhumuza Kananura, PhD

Academic Editor

PLOS ONE

Additional Editor Comments (optional):

This statement in the abstract needs to be revised to have it aligned with how the study contributes to the current literature of understanding the consistent high stillbirths in low-resource settings of SSA or Ghana.  

"Published data on stillbirths and their associating factors in the Volta Region of Ghana is limited and adversely affects a good understanding of local factors that must be considered in designing appropriate interventions." 

Reviewers' comments:

Reviewer's Responses to Questions

**Comments to the Author**

1. If the authors have adequately addressed your comments raised in a previous round of review and you feel that this manuscript is now acceptable for publication, you may indicate that here to bypass the “Comments to the Author” section, enter your conflict of interest statement in the “Confidential to Editor” section, and submit your "Accept" recommendation.

Reviewer #1: All comments have been addressed

Reviewer #2: All comments have been addressed

2. Is the manuscript technically sound, and do the data support the conclusions?

Reviewer #1: Yes

Reviewer #2: Yes

3. Has the statistical analysis been performed appropriately and rigorously? 

Reviewer #1: Yes

Reviewer #2: Yes

4. Have the authors made all data underlying the findings in their manuscript fully available?

Reviewer #1: Yes

Reviewer #2: Yes

5. Is the manuscript presented in an intelligible fashion and written in standard English?

Reviewer #1: Yes

Reviewer #2: Yes

6. Review Comments to the Author

Reviewer #1: The authors have satisfactorily attended to comments from the initial review. I have no further comments

Reviewer #2: based in the responses received, the authors have answered all queries that were raised. The paper is not acceptable in the current form

7. PLOS authors have the option to publish the peer review history of their article (what does this mean?). If published, this will include your full peer review and any attached files.

Reviewer #1: **Yes: **Addmore Chadambuka

Reviewer #2: **Yes: **Christian Obirikorang

---

## [Editor Report · Acceptance letter]

11 Dec 2023

PONE-D-23-17720R1 

Stillbirth incidence and determinants in a tertiary health facility in the Volta Region of Ghana. 

Dear Dr. Osarfo:

I'm pleased to inform you that your manuscript has been deemed suitable for publication in PLOS ONE. Congratulations! Your manuscript is now with our production department. 

Kind regards, 

on behalf of

Dr. Rornald Muhumuza Kananura 

Academic Editor

PLOS ONE